# Efficacy and Safety of Visible and Near-Infrared Photobiomodulation Therapy on Astenospermic Human Sperm: Wavelength-Dependent Regulation of Nitric Oxide Levels and Mitochondrial Energetics

**DOI:** 10.3390/biology14050491

**Published:** 2025-05-01

**Authors:** Matilde Balbi, Rachele Lai, Sara Stigliani, Claudia Massarotti, Matteo Bozzo, Paola Scaruffi, Silvia Ravera, Andrea Amaroli

**Affiliations:** 1Experimental Medicine Department, University of Genova, 16132 Genova, Italy; matilde.balbi@unige.it (M.B.); silvia.ravera@unige.it (S.R.); 2BIO-Photonics Overarching Research Laboratory, Department of Earth, Environmental and Life Sciences (DISTAV), University of Genova, 16132 Genova, Italy; lai.rachele.98@gmail.com (R.L.); matteo.bozzo@unige.it (M.B.); 3SSD Physiopathology of Human Reproduction, IRCCS Ospedale Policlinico San Martino, 16132 Genova, Italy; sara.stigliani@hsanmartino.it (S.S.); claudia.massarotti@unige.it (C.M.); 4Department of Neuroscience, Rehabilitation, Ophthalmology, Genetics and Maternal-Child Health (DiNOGMI), University of Genova, 16132 Genova, Italy; 5IRCCS Ospedale Policlinico San Martino, 16132 Genova, Italy; 6Interuniversity Center for the Promotion of the 3Rs Principles in Teaching and Research (Centro 3R), 56122 Pisa, Italy

**Keywords:** infertility, asthenozoospermia, sperm motility, mitochondrial metabolism, DNA fragmentation, male fertility, oxidative stress, low-level laser therapy

## Abstract

Male infertility is a growing concern, with many men experiencing reduced sperm motility, a condition known as asthenozoospermia. Healthy sperm must progressively move to reach and fertilize an oocyte. This ability depends on the energy produced by tiny structures inside the sperm called mitochondria. Another critical factor is nitric oxide, a molecule that can support or damage sperm function depending on its levels. This study explored whether exposure to specific types of light, known as photobiomodulation therapy, could improve sperm energy production and movement. Semen samples were exposed for 60 s to light wavelengths and parameters, and their energy ratio, oxidative stress damage, and nitric oxide levels were measured. The results showed that most visible and near-infrared light significantly increased sperm energy without causing damage, while light at 635 nm reduced energy production. These data indicate that light therapy, when applied with appropriate parameters, may offer a promising approach to improve sperm function in men experiencing fertility problems. These findings provide new insights into how light interacts with sperm cells and may lead to future treatments that enhance male fertility safely and effectively.

## 1. Introduction

Over the past decades, numerous studies have highlighted a progressive decline in sperm quality, with significant implications for male fertility. Levine et al. [1,2] documented a 50% reduction in sperm quality over the last 40 years, underscoring a worrying global trend. Similarly, Agarwal et al. [3] analyzed the prevalence of male infertility, revealing a considerable geographic variability in its distribution. Male infertility is now recognized as a global public health issue with a complex and multifactorial etiology. Contributing factors include genetic abnormalities, hormonal imbalances, reproductive tract obstructions, lifestyle-related factors, and environmental exposures, all of which can impair both spermatogenesis—the process of sperm cell production—and the ability of sperm to fertilize the oocyte [4,5,6].

Mitochondrial dysfunction and alterations in nitric oxide (NO) metabolism play a central role in impairments of sperm function [7,8,9,10,11]. During spermatogenesis, mitochondria undergo significant morphological and functional changes. Mitochondria show low oxidative phosphorylation (OxPhos) activity in spermatogonia and immature germ cells. In contrast, mature spermatozoa form a mitochondrial sheath in the flagellar region, essential for motility. These organelles produce ATP via the electron transport chain, involving respiring complexes I-IV and ATP synthase. The energy generated is crucial for sperm motility, providing the propulsion necessary to reach and fertilize the oocyte [12,13].

Nitric oxide (NO) also plays a pivotal role in regulating spermatogenesis by modulating Sertoli and germ cells’ activity and influencing sperm maturation. At low concentrations, NO activates the cyclic guanosine monophosphate (cGMP) signaling pathway, promoting sperm motility [14].

Thus, alterations in OxPhos function and NO metabolism—caused by mutations, oxidative/nitrosative stress, and environmental factors—can compromise sperm energy efficiency, structure, and physiological functionality [8,11,15]. Indeed, the excessive production of reactive oxygen species (ROS) in dysfunctional mitochondria can damage DNA, cellular membranes, and proteins [16,17], leading to an energy deficit and impaired sperm motility [18,19]. ATP generation is essential for proper flagellar movement, regulating microtubule structure and dynein interaction [12]. Moreover, although NO plays a physiological role in maintaining sperm function, excessive concentration can contribute to sperm damage, ultimately reducing sperm quality, particularly in terms of morphology and motility [20].

In a recent study [21], we demonstrated that photobiomodulation (PBM) treatment at 810 nm, 1 W, 60 J/cm^2^ (60 s—spot size 1 cm^2^) significantly improved the motility of sperm samples collected from 70 asthenozoospermic patients. A progressive motility increase was observed immediately after irradiation, with elevated levels maintained up to 30 min post-treatment, followed by a decline at 60 min, returning to near-control values. Additionally, treatment significantly reduced the immotile sperm fraction and improved sperm viability over time. The mechanism of action of laser therapy in asthenozoospermic spermatozoa was based on photon-induced stimulation of mitochondrial activity, leading to a significant increase in ATP production. It enhanced aerobic oxidation while preserving cell membrane integrity.

Similar findings have been reported by other research groups, highlighting the potential of PBM in treating male infertility [22,23,24,25,26]. Different wavelengths within the visible and near-infrared (NIR) spectrum, combined with specific energy parameters, have been shown to affect sperm motility [22,23,24,27,28,29,30,31,32,33,34,35,36,37,38,39,40,41,42,43]. Beyond sperm cells, photons with wavelengths ranging from visible to NIR (up to approximately 1000 nm) have been demonstrated to photo-energize mitochondria, modulating the activity of the respiratory chain and cellular metabolism [44,45,46,47,48,49,50,51]. Photobiomodulation has also been described as a potential modulator of NO metabolism, particularly in conditions of endothelial dysfunction [52]. In particular, wavelength-dependent vasodilation and NO release have been noted to be more effectively induced by red light than near-infrared wavelengths [53]. However, disruptions in NO homeostasis and OxPhos can impair proper mitochondrial coupling, increasing oxidative stress and negatively affecting cellular physiology [54,55].

Given the gap of consistent information on the relationship between PBM and the restoration of mitochondrial activity in asthenozoospermic spermatozoa, we conducted a study to assess the influence of visible and NIR light at wavelengths of 450 nm, 635 nm, 810 nm, 940 nm, and 1064 nm, exposing a spot-size area of 1 cm^2^ to variable power levels from low to higher (0.25 W, 0.5 W, 1 W, 2 W) for 60 s. This study aimed to investigate the effects of a broad spectrum of wavelengths and therapeutic parameters to provide scientific evidence on the efficacy and safety of PBM for treating asthenozoospermia.

## 2. Materials and Methods

### 2.1. Patient Recruitment and Experimental Design

Semen samples were obtained from donors undergoing semen analysis for fertility diagnosis at the SSD Physiopathology of Human Reproduction of the IRCCS Ospedale Policlinico San Martino, Genoa, Italy, from January 2024 to January 2025. This study complies with the Declaration of Helsinki [56] and was approved by the Liguria Regional Ethics Committee (219/2024—DB id 13852). All donors provided written informed consent, and participation was voluntary. All patients collected seminal fluid via masturbation into a sterile container after 3–5 days of abstinence. The seminal fluid samples were liquefied at room temperature for 30 min. Semen analysis was conducted according to the World Health Organization (WHO) criteria [57], evaluating the following parameters: semen volume, sperm concentration per milliliter, and sperm motility. To ensure maximum reliability of the results, the diagnosis of asthenozoospermia was confirmed through two independent blinded evaluations conducted by experienced seminologists. In the event of discrepancies between the two assessments, a third experienced healthcare operator conducted an additional analysis to determine the definitive diagnosis. The study inclusion criteria involved selecting patients with asthenozoospermia, defined as total motility below 42% and progressive motility below 30%. Samples with a seminal volume of less than 1.5 mL, sperm agglutination, round or amorphous cells, bacterial contamination, oligospermia, or genetic syndromes were excluded from the study. Applying the inclusion and exclusion criteria, out of 75 patients who presented as donors and underwent semen analysis for fertility diagnosis, 43 were classified as asthenozoospermic. Of these, 20 were included in the study, while 23 were excluded due to bacterial contamination of the sample or inadequate volume for investigation. The demographic data of the 20 patients enrolled in the study and their semen parameters before the experiments are shown in Table 1.

Photobiomodulation treatments were performed on native ejaculates. The samples were irradiated for 60 s at different wavelengths and power levels, as detailed in Section 2.2:450 nm at 0.25, 0.50, 1.00, and 2.00 W; control at 0 W635 nm at 0.25, 0.50, 1.00, and 2.00 W; control at 0 W810 nm at 0.25, 0.50, 1.00, and 2.00 W; control at 0 W940 nm at 0.25, 0.50, 1.00, and 2.00 W; control at 0 W1064 nm at 0.25, 0.50, 1.00, and 2.00 W; control at 0 W

After irradiation, some aliquots were maintained at 37 °C for 10 min and then frozen at −80 °C. Other aliquots were irradiated with the same wavelengths and parameters but kept at 37 °C for 60 min before freezing at −80 °C.

The resulting samples were analyzed to observe potential effects on cellular metabolism, oxidative stress damage, and nitric oxide levels. Figure 1 summarizes the experimental setup.

### 2.2. Technical Specifications of the Equipment Utilized for Spermatozoa Irradiation

The irradiations were conducted using the ENEA Trio diode laser system (manufactured by Garda Laser S.A.S., Verona, Italy). This advanced device could emit laser light at precise wavelengths of 450 nm, 635 nm, 810 nm, 940 nm, and 1064 nm (±2 nm tolerance) in continuous wave mode, with each irradiation session lasting 60 s. The power output was adjustable to 0.25 W, 0.50 W, 1.00 W, or 2.00 W, corresponding to energy doses of 15.0 J, 30.0 J, 60.0 J, or 120.0 J, respectively. This resulted in a power density range from 0.25 W/cm^2^ to 2.00 W/cm^2^ and a fluence range from 15.0 J/cm^2^ to 120.0 J/cm^2^.

To simplify, the therapies will be abbreviated in the text as 0.25 W (15 J), 0.50 W (30 J), 1.00 W (60 J), and 2.00 W (120 J).

Control samples were subjected to a power setting of 0.0 W to ensure baseline comparisons.

A flat-profiled handpiece (FT-HP) was employed to deliver the irradiation. Prior characterization of the FT-HP demonstrated its ability to provide a uniform and stable energy distribution across a spot area of 1 cm^2^, regardless of the distance from the target surface, as documented in previous studies [58,59]. To facilitate precise targeting during near-infrared irradiations, a 635 nm red light pointer (with negligible power output, <0.5 mW) was utilized to visualize the exposed area, ensuring experimental accuracy and maintaining blinding protocols. To avoid potential biases, the experiments were performed without any artificial light. Table 2 summarizes the instruments used for irradiation and the parameters delivered.

The accuracy of the laser parameters was verified using a Pronto-250 power meter (Gentec Electro-Optics, Inc., G2E Quebec City, QC, Canada). The FT-HP was securely mounted on a stand and operated in contact mode with the surface of a multiwell plate, where each well had a diameter of approximately 1.1 cm [47]. To minimize potential interference from light reflections, the multiwell plate was placed on a Metal Velvet light-absorbing plate (Acktar Ltd., Kiryat-Gat, Israel). These plates are designed to exhibit extremely low reflectance, guaranteed to be ≤1% in the near-infrared spectrum, thereby significantly reducing unwanted reflections from the work surface [47]. Thermal effects were meticulously monitored using an OMEGA thermal monitoring system with the following specifications: operating range from −200 to 390 °C, resolution (internal and displayed) of 0.5 °C (1 °F), and accuracy of ±0.5 °C (model: OM-EL-USB-TC-LCD, OMEGA Engineering Limited, M44 5BD Manchester, UK).

### 2.3. Detection of Adenosine Triphosphate in Spermatozoa

The ATP concentration was measured using specific enzymatic assays by monitoring NADP reduction at 340 nm [60]. Irradiated sperm samples and corresponding controls were added to a reaction mixture containing 50 mM Tris-HCl (pH 8.0), 1 mM NADP, 0.5 mM MgCl_2_, and 5 mM glucose. Subsequently, a purified mixture of hexokinase and glucose-6-phosphate dehydrogenase was added. Protein quantification was performed using the Bradford method [61].

### 2.4. Detection of Adenosine Monophosphate in Spermatozoa

AMP concentration was assayed using specific enzymatic assays by monitoring NADH oxidation [62]. The assay medium contained 50 mM Tris-HCl (pH 8.0), 0.2 mM NADH, 0.5 mM MgCl_2_, 1 mM phosphoenolpyruvate, 0.2 mM ATP, and 5 μg of a purified mixture of pyruvate kinase and lactate dehydrogenase. Subsequently, 2 μg of purified adenylate kinase and 2 μg of kinase plus lactate dehydrogenase were added. Protein quantification was performed using the Bradford method [61].

### 2.5. Detection of Oxidative Stress Damage in Spermatozoa

#### 2.5.1. Malondialdehyde Assay

For the determination of malondialdehyde (MDA), the thiobarbituric acid reactive substances (TBARS) assay [63] was used, with minor modifications. This method is based on the reaction of MDA, a degradation product of lipid peroxides, with thiobarbituric acid (TBA). The TBARS solution was prepared with 15% trichloroacetic acid (TCA) in 0.25 N HCl and 26 mM TBA. To assess the baseline MDA concentration, 600 μL of the TBARS solution was added to 50 μg of total proteins dissolved in 300 μL of Milli-Q water. The mixture was incubated for 45 min at 100 °C, followed by centrifugation at 14,000 rpm for 2 min. The supernatant was then analyzed spectrophotometrically at a wavelength of 532 nm.

#### 2.5.2. 8-Hydroxy-2′-deoxyguanosine Assay

The quantification of 8-hydroxy-2′-deoxyguanosine (8-OHdG) was assessed using the high-sensitivity ELISA assay (Kit-code: ab201734 from Abcam plc, Discovery Drive, Cambridge Biomedical Campus, Cambridge, UK) [64].

### 2.6. Detection of Nitric Oxide in Spermatozoa

Nitric oxide was quantified by measuring nitrite with the Griess reagent using the Nitrite Assay Kit (Catalog No.: 23479-1KT-F, Sigma-Aldrich, Merck KGaA, Frankfurter Str. 250, 64293 Darmstadt, Germany) [65].

### 2.7. Statistical Analysis

Following the previous study by Stigliani et al. [21], the sample size was determined based on preliminary data, considering a beta value of 0.2, a significance level (alpha) of 0.05, and a statistical power of 0.8, resulting in a minimum of 18 patients in total [66].

Data normality was assessed using the Shapiro–Wilk test (*p* > 0.05 indicating normal distribution). Specifically, the observed p-values ranged from 0.08 to 0.92, supporting parametric tests. Data analysis of means of the single measures was conducted using one-way analysis of variance (ANOVA) followed by Sidak’s multiple comparisons test to adjust for multiple testing and control the family-wise error rate. Statistical processing was performed using Prism 10 software (GraphPad Software, Boston, MA, USA), with statistical significance set at *p* < 0.05.

## 3. Results

### 3.1. ATP Evaluation in Spermatozoa in Response to Photobiomodulation

ATP is the primary energy source for cellular functions. We investigated potential variations in ATP concentration in spermatozoa following irradiation with PBM to determine whether these changes persist up to 60 min post-treatment or stabilize over time. ATP concentration in spermatozoa was measured 10 min and 60 min after PBM irradiation using laser light at different wavelengths (450 nm, 635 nm, 810 nm, 940 nm, and 1064 nm) with power outputs of 0.25 W, 0.50 W, 1.00 W, and 2.00 W, for 60 s, with a spot size of 1 cm^2^.

The results indicate a variable effect depending on wavelength and power, with significant increases or decreases in ATP concentration.

Figure 2 reports the ATP concentration in spermatozoa 10 min post-irradiation.

Regarding the effect of PBM at 450 nm, no significant changes in ATP concentration were observed at 0.25 W (15 J) (*p* > 0.05). However, a significant increase in ATP concentration was noted at 0.50 W (30 J) and 1.00 W (60 J) (*p* < 0.001), while no significant changes were observed at 2.00 W (120 J) (*p* > 0.05). For PBM at 635 nm, a significant decrease in ATP concentration was recorded across all tested power outputs (*p* < 0.001), with the most significant reduction observed at 2.00 W (120 J). At 810 nm, a significant increase in ATP concentration was induced by 0.25 W (15 J), 0.50 W (30 J), and 1.00 W (60 J) (*p* < 0.001), with no significant changes at 2.00 W (120 J) (*p* > 0.05). Irradiation with 1.00 W (60 J) is the treatment that induces the greatest increase in ATP at 10 min after treatment. At 940 nm, no significant variations were observed at 0.25 W (15 J) and 2.00 W (120 J) (*p* > 0.05), while significant increases were noted at 0.50 W (30 J) and 1.00 W (60 J) (*p* < 0.001). Finally, at 1064 nm, a significant increase in ATP concentration was observed at 0.25 W (15 J) and 0.50 W (30 J) (*p* < 0.001), while 1.00 W (60 J) and 2.00 W (120 J) resulted in a significant reduction (*p* < 0.001).

Figure 3 illustrates the ATP concentration in spermatozoa 60 min post-treatment.

At 450 nm, no significant changes were observed at 0.25 W (15 J) (*p* > 0.05), while ATP concentration increased at 0.50 W (30 J) and 1.00 W (60 J) (*p* < 0.001). At 2.00 W (120 J), no modulation of ATP concentration was observed (*p* > 0.05). At 635 nm, all treatments induced a significant reduction in ATP concentration (*p* < 0.001), with the most significant decrease at 2.00 W (120 J). At 810 nm, irradiation at 0.25 W (15 J), 0.50 W (30 J), and 1.00 W (60 J) produced a significant increase in ATP concentration (*p* < 0.001), while no significant changes were observed at 2.00 W (120 J) (*p* > 0.05). At 940 nm, no significant variations were observed at 0.25 W (15 J) and 2.00 W (120 J) (*p* > 0.05), while significant increases were noted at 0.50 W (30 J) and 1.00 W (60 J) (*p* < 0.001). At 1064 nm, a significant increase in ATP was observed at 0.25 W (15 J) and 0.50 W (30 J) (*p* < 0.001), while 1.00 W (60 J) and 2.00 W (120 J) caused significant inhibition (*p* < 0.001). Comparison of the evaluations at 10 and 60 min after treatment showed a decreasing trend in ATP concentration in controls and treated independently of the stimulating or inhibiting effect (*p* < 0.01–*p* < 0.001).

### 3.2. AMP Evaluation in Spermatozoa in Response to Photobiomodulation

AMP is a key indicator of cellular energy status and mitochondrial metabolism regulation. AMP concentration increases when cellular energy levels are reduced, signaling low energy availability. AMP concentration in spermatozoa was measured 10 min and 60 min after PBM irradiation using laser light at different wavelengths (450 nm, 635 nm, 810 nm, 940 nm, and 1064 nm) with power outputs of 0.25 W, 0.50 W, 1.00 W, and 2.00 W, for 60 s, with a spot size of 1 cm^2^. As for ATP content, the results indicate a variable effect depending on wavelength and power.

Figure 4 illustrates the AMP concentration in spermatozoa 10 min post-irradiation.

At 450 nm, no significant changes were observed at 0.25 W (15 J) (*p* > 0.05), but a significant reduction in AMP concentration was recorded at 0.50 W (30 J) and 1.00 W (60 J) (*p* < 0.001), while no significant changes were observed at 2.00 W (120 J) (*p* > 0.05). At 635 nm, a significant increase in AMP concentration was observed at 0.50 W (30 J), 1.00 W (60 J), and 2.00 W (120 J) (*p* < 0.001). At 810 nm, all irradiations produced a significant decrease in AMP concentration (*p* < 0.001), with the most significant change at 1.00 W (60 J). At 940 nm, no significant variations were observed at 0.25 W (15 J) and 2.00 W (120 J) (*p* > 0.05), but a significant decrease was noted at 0.50 W (30 J) (*p* < 0.01) and 1.00 W (60 J) (*p* < 0.001). At 1064 nm, a significant decrease in AMP concentration was observed at 0.25 W (15 J) and 0.50 W (30 J) (*p* < 0.001), while a significant increase was induced at 2.00 W (120 J) (*p* < 0.01).

Figure 5 illustrates the AMP concentration in spermatozoa 60 min post-treatment.

At 450 nm, no significant changes were observed under any conditions (*p* > 0.05). At 635 nm, a significant increase in AMP concentration was observed at 0.25 W (15 J) (*p* < 0.01). At the same time, all other power outputs produced a significant increment (*p* < 0.001), with the greatest increase at 2.00 W (120 J). At 810 nm, AMP concentration decreased significantly at 0.25 W (15 J) (*p* < 0.01), while it decreased significantly at other power outputs, with the most significant decrement at 1.00 W (60 J) (*p* < 0.001). At 940 nm, no significant variations were observed under any conditions (*p* > 0.05). Finally, at 1064 nm, a significant decrease was observed at 0.25 W (15 J) and 0.50 W (30 J) (*p* < 0.001), while 1.00 W (60 J) and 2.00 W (120 J) induced a significant increase in AMP concentration (*p* < 0.001), with the most significant increase recorded at 2.00 W (120 J). Irradiation with 810 nm at all irradiation parameters and 1064 nm at 0.25 W (15 J) and 0.50 W (30 J) induced the greatest decrease in AMP concentration 10 min after treatment (*p* < 0.01). In contrast, 635 nm at 0.50 W (30 J), 1.00 W (60 J), and 2.00 W (120 J) induced the greatest increase (*p* < 0.01). Comparison of the measurements at 10 and 60 min after treatment showed that the AMP concentration decreased over time in control and treated samples (*p* < 0.01–*p* < 0.001).

### 3.3. Energy Status (ATP/AMP Ratio) Evaluation in Spermatozoa in Response to Photobiomodulation

The ATP/AMP ratio is a key indicator of cellular energy balance. Under normal conditions, cells maintain high ATP levels and low AMP levels. However, ATP is degraded under energy stress to adenosine diphosphate (ADP) and AMP. An increase in AMP concentration may signal energy depletion. The energy ratio was calculated based on ATP and AMP concentrations measured 10 min and 60 min after PBM irradiation using laser light at different wavelengths (450 nm, 635 nm, 810 nm, 940 nm, and 1064 nm) with power outputs of 0.25 W, 0.50 W, 1.00 W, and 2.00 W, for 60 s, with a spot size of 1 cm^2^.

Figure 6 shows the ATP/AMP ratio in spermatozoa 10 min post-irradiation.

At 450 nm, a significant increase in the ATP/AMP ratio was observed at 0.50 W (30 J) and 1.00 W (60 J) (*p* < 0.001), while no significant changes were observed at 0.25 W (15 J) and 2.00 W (120 J) (*p* > 0.05). At 635 nm, a more evident and significant decrease in the ATP/AMP ratio was observed at 0.50 W (30 J), 1.00 W (60 J), and 2.00 W (120 J) (*p* < 0.01, and *p* < 0.001, respectively). At 810 nm, all irradiations produced a significant increase in the ATP/AMP ratio (*p* < 0.001), with the most significant change at 1.00 W (60 J). Irradiation at 1.00 W (60 J) induced the highest ATP/AMP score at 10 min after treatment. At 940 nm, a significant increase was observed at 0.50 W (30 J) and 1.00 W (60 J) (*p* < 0.001), while no significant changes were observed at 0.25 W (15 J) and 2.00 W (120 J) (*p* > 0.05). At 1064 nm, the ATP/AMP ratio increased significantly at 0.25 W (15 J) and 0.50 W (30 J) (*p* < 0.001), while a significant decrease was observed at 1.00 W (60 J) (*p* < 0.05) and 2.00 W (120 J) (*p* < 0.05).

Figure 7 reports the ATP/AMP ratio in spermatozoa 60 min post-irradiation.

At 450 nm, a significant increase in the ATP/AMP ratio was observed at 0.50 W (30 J) and 1.00 W (60 J) (*p* < 0.001), while no significant changes were observed at 0.25 W (15 J) and 2.00 W (120 J) (*p* > 0.05). At 635 nm, a significant decrease in the ATP/AMP ratio was observed across all parameters (0.25 W, 0.50 W, 1.00 W, and 2.00 W) (*p* < 0.001). At 810 nm, a significant increase in the ATP/AMP ratio was recorded across all irradiation doses (*p* < 0.001), with the most significant change at 1.00 W (60 J). At 940 nm, a significant increase was observed at 0.50 W (30 J) and 1.00 W (60 J) (*p* < 0.01 and *p* < 0.001), while no significant changes were observed at 0.25 W (15 J) and 2.00 W (120 J) (*p* > 0.05). At 1064 nm, a significant increase in the ATP/AMP ratio was observed at 0.25 W (15 J) and 0.50 W (30 J) (*p* < 0.001), while a decrease was observed at 1.00 W (60 J) and 2.00 W (120 J) (*p* < 0.001). A comparison of the evaluations at 10 and 60 min after treatment showed a decreasing trend in the ATP/AMP ratio in the control group, while the treated groups exhibited a similar trend—except for the treatments at 1064 nm with 1.00 W (60 J) and 2.00 W (120 J), where the ratio remained constant.

### 3.4. Evaluation of Oxidative Stress Damage in Spermatozoa in Response to Photobiomodulation

Damage to cellular membranes and DNA resulting from oxidative stress was evaluated 10 min and 60 min after PBM irradiation using laser light at different wavelengths (450 nm, 635 nm, 810 nm, 940 nm, and 1064 nm) with power outputs of 0.25 W, 0.50 W, 1.00 W, and 2.00 W, for 60 s, with a spot size of 1 cm^2^.

#### 3.4.1. Evaluation of Malondialdehyde

MDA is a marker of oxidative stress damage to cellular membranes and is a lipid peroxidation product. Data analysis shown in Figure 8 and Figure 9 revealed no statistically significant differences between the treated groups and the control group under any experimental conditions (*p* > 0.05), indicating that PBM did not significantly influence MDA levels. A comparison of measurements at 10 and 60 min after treatment showed that the MDA concentration did not decrease over time in the control and treated samples (*p* > 0.05).

#### 3.4.2. Evaluation of 8-Hydroxy-2′-Deoxyguanosine

8-OHdG is a biomarker used to measure DNA damage resulting from oxidative stress. It is a modified form of deoxyguanosine, one of the DNA bases, formed when DNA is damaged by ROS. Specifically, 8-OHdG is one of the primary products of purine base oxidation, and its concentration in cells is used as an indicator of oxidative damage at the genetic level. Data analysis shown in Figure 10 and Figure 11 revealed no statistically significant differences between the treated groups and the control group under any experimental conditions (*p* > 0.05), indicating that PBM did not significantly influence 8-OHdG levels. A comparison of measurements at 10 and 60 min after treatment showed that the 8-OHdG concentration did not decrease over time in the control and treated samples (*p* > 0.05).

### 3.5. Evaluation of Nitric Oxide in Spermatozoa in Response to Photobiomodulation

Nitric oxide is fundamental in various biological processes, including regulating sperm motility and fertilizing capacity. Its action is highly concentration-dependent: at physiological levels, NO exerts beneficial effects on mitochondrial activity, whereas at elevated concentrations, it can impair sperm functionality.

We assessed NO production by measuring nitrite concentrations, which are metabolic byproducts of nitric oxide. Measurements were performed at 10 and 60 min following laser light irradiation at different wavelengths (450 nm, 635 nm, 810 nm, 940 nm, and 1064 nm) and varying power levels (0.25 W, 0.50 W, 1.00 W, and 2.00 W). The irradiation was applied for 60 *s* over an exposure area of 1 cm^2^.

Figure 12 reports the results of NO concentration in spermatozoa 10 min after irradiation.

At 450 nm, a significant increase in NO levels was observed at 0.50 W (30 J) and 1.00 W (60 J) (*p* < 0.001), whereas no significant differences were detected at 0.25 W (15 J) and 2.00 W (120 J) compared to the control group (*p* > 0.05). At 635 nm, NO concentration was significantly elevated at all tested power levels (*p* < 0.001). At 810 nm, all power levels induced a significant increase in NO concentration, particularly at 0.25 W (15 J) (*p* < 0.001), 0.50 W (30 J), and 2.00 W (120 J) (*p* < 0.001), with the highest increase observed at 1.00 W (60 J) (*p* < 0.001). At 940 nm, a significant increase in NO levels was observed at all power levels: 0.25 W (15 J) (*p* < 0.001), 0.50 W (30 J), and 1.00 W (60 J) (*p* < 0.001), while the increase at 2.00 W (120 J) was less pronounced (*p* < 0.01). Finally, at 1064 nm, NO levels were significantly elevated at all examined power levels: 0.25 W (15 J) (*p* < 0.001), 0.50 W (30 J), 1.00 W (60 J), and 2.00 W (120 J) (*p* < 0.001).

Figure 13 illustrates the NO concentration in spermatozoa 60 min after laser irradiation.

At 450 nm, a significant increase in NO levels was detected only at 0.50 W (30 J) (*p* < 0.05) and 1.00 W (60 J) (*p* < 0.001), whereas no significant variations were observed at 0.25 W (15 J) and 2.00 W (120 J) compared to the control group (*p* > 0.05). At 635 nm, NO concentration remained significantly elevated under all experimental conditions (*p* < 0.001). At 810 nm, a significant increase was recorded at 0.50 W (30 J) and 2.00 W (120 J) (*p* < 0.001), with the highest increase at 1.00 W (60 J) (*p* < 0.001). However, at 0.25 W (15 J), no significant differences were observed (*p* > 0.05). At 940 nm, NO concentration significantly increased at 0.25 W (15 J) (*p* < 0.05), 0.50 W (30 J), and 1.00 W (60 J) (*p* < 0.001), whereas at 2.00 W (120 J), no significant changes were detected compared to the control (*p* > 0.05). Finally, at 1064 nm, a significant increase was observed at 0.50 W (30 J) (*p* < 0.01), 1.00 W (60 J), and 2.00 W (120 J) (*p* < 0.001).

At 60 min, NO levels remained elevated compared to control groups; however, a significant reduction was observed relative to values recorded at 10 min.

## 4. Discussion

Ejaculated spermatozoa are highly specialized cells, whose role is to reach and fertilize the oocyte. During spermatogenesis, spermatic DNA undergoes extreme compaction by replacing histones with protamines, which makes the genetic material almost inaccessible to transcription. The absence of functional ribosomes and active endoplasmic reticulum excludes the possibility of protein translation after spermatozoon maturation [67,68]. Consequently, the regulation of spermatic activity primarily depends on post-translational biochemical modifications, including capacitation, phosphorylation, and the activation of second messengers such as cAMP and its energetics [69,70]. The energy produced by spermatozoa is mainly used to support flagellar motility, with energy metabolism adapting to the environmental conditions of the female reproductive tract. Depending on substrate and oxygen availability, spermatozoa alternate between glycolysis and OxPhos [13,70]. Alterations in mitochondrial function are often associated with spermatic impairments and reduced fertilizing capacity [12,71].

A recent study by Stigliani et al. [21] has shown a significant reduction in mitochondrial activity and an increase in oxidative stress in spermatozoa of asthenozoospermic patients compared to normozoospermic men. The sperm energy deficit was partially compensated through PBM using an 810 nm diode laser, delivered at a power of 1 W and an energy of 60 J for 60 s over an area of 1 cm^2^. The metabolic changes induced by PBM were closely correlated with improvements in sperm motility, mainly through activation of the OxPhos, while glycolysis showed no significant variations. This observation is consistent with the fact that the glycolytic pathway, lacking photoacceptive molecules, is not a direct target of light, unlike the mitochondrial chromophores involved in OxPhos [21]. In ciliated cell models, a strong correlation has been demonstrated between PBM stimulus, increased mitochondrial activity [51], and accelerated ciliary movement during swimming [72]. Furthermore, evidence from the literature indicates that the range of visible and NIR light can modulate sperm motility, even hours after treatment [23,27,29,37,73,74,75,76,77,78,79].

Our data confirm that wavelengths within the visible and NIR range can modulate sperm energy metabolism without inducing oxidative damage. In line with the observations by Stigliani et al. [21], these effects are rapid and evident as early as 10 min after irradiation. The observed metabolic changes depend on the wavelength used and the energy delivered. Spermatozoa treated at 810 nm, along with those irradiated at 450 nm and 940 nm at 0.50 W (30 J) and 1.00 W (60 J), and 1064 nm at 0.25 W (15 J) and 0.50 W (30 J), show a significant increase in ATP levels compared to controls. Conversely, irradiation at 635 nm results in a marked reduction in ATP levels at all tested parameters, as observed with higher-intensity irradiation at 1064 nm. Notably, although the effectiveness of PBM on energy metabolism varies and is modulated by wavelength spectrum and administration parameters, the safety of treatment in terms of oxidative stress damage appears consistent across all experimental conditions without causing an increase in lipid or DNA damage levels. ATP in spermatozoa is essential for dynein activity, the motor protein responsible for flagellar movement [80,81]. During this process, ATP is progressively hydrolyzed into ADP and subsequently into AMP by adenylate kinase and ATPase enzymes. Therefore, the measurement of AMP levels and calculation of the ATP/AMP ratio represent the energy status of spermatozoa in response to PBM.

Our results suggest that, except for some wavelengths at extreme parameters [0.25 W (15 J) and 2 W (120 J)], PBM in the visible and NIR light range induces an energy increase in asthenozoospermic spermatozoa compared to untreated controls. In particular, irradiation at 810 nm was the most effective wavelength, capable of positively affecting a wide range of parameters without window or hormetic effects. In contrast, treatments at 635 nm negatively affected sperm energetics. The 1064 nm wavelength showed a biphasic behavior, with beneficial effects at low intensities [0.25 W (15 J) and 0.5 W (30 J)], like those of 810 nm, while at higher powers it induced an energy reduction like that observed at 635 nm. This suggests a complex interaction with spermatic mitochondrial metabolism.

Numerous components, including flavins, heme groups, Cu_A and Cu_B centers, lipids, and water, can produce optical absorption, as described by the Lambert–Beer law [46,82,83,84,85]. Flavins absorb in blue light (including our 450 nm), heme groups strongly absorb in the visible range (blue and red, thus including our 450 and 635 nm), while the Cu_A center has significant absorption around 830 nm [86,87,88]. Lipids [83] and water [89] absorb in the NIR range, starting from our 940 nm. Therefore, while the effects in the range of 450 nm to 810 nm are compatible with photo-energization of mitochondrial complexes by electronic transitions, at longer wavelengths, the interaction with water and lipids may cause photothermal effects at macro- and nanoscales. This phenomenon may lead to changes in the viscosity and fluidity of cell membranes, including the mitochondrial cristae where OxPhos complexes are embedded or adjacent. Santana-Blank and Sommer [84,85,90] hypothesize that PBM effects may involve water resonance, which can lead to changes in membrane properties and a faster ATP synthase function in ATP production.

After 60 min of irradiation and incubation at 37 °C, the energetics of asthenozoospermic spermatozoa follow the trend observed at 10 min. However, the profile shows a predictable physiological decline in the ATP/AMP ratio, which was also observed in the controls. This decline is consistent with what has been observed in isolated mitochondria, where after an immediate stimulation of the OxPhos complex correlated with increased oxygen consumption and ATP production, there is a gradual return to the control energy conditions [47]. In asthenozoospermic spermatozoa, the decline appeared faster in treated samples compared to controls, likely due to the faster depletion of metabolic resources in an isolated environment such as ejaculated semen. Conversely, it is noteworthy that at 635 nm, the percentage of the inhibition of the energy ratio remained constant over time, around 50% compared to the control.

The high energy profile of asthenozoospermic spermatozoa at 60 min, although reduced compared to 10 min, highlights a relatively lasting effect of PBM on energy metabolism. This aspect has fundamental importance considering the key role of ATP in critical processes such as the fusion of the sperm plasma membrane with the acrosomal membrane and the subsequent acrosomal reaction, essential for sperm penetration into the oocyte [91,92,93]. Moreover, ATP is essential for maintaining cellular homeostasis, ion balance, and membrane potential, which ensure sperm survival and functionality in the fertilization process [94].

Nitric oxide, along with ATP, plays a fundamental role in regulating the motility and fertilizing capacity of human ejaculated spermatozoa [95,96,97]. Its complex and multifactorial action contributes to motility regulation by activating the cyclic guanosine monophosphate (cGMP) signaling pathway and influencing intracellular cyclic adenosine monophosphate (cAMP) levels. Additionally, NO is involved in capacitation and acrosomal reaction processes and directly modulates mitochondrial OxPhos activity [98].

In irradiated asthenozoospermic spermatozoa, NO production showed a trend consistent with variations in cellular energetics up to the 810 nm wavelength. Irradiation at 450 nm with powers of 0.25 W (15 J) and 2 W (120 J) did not induce significant changes in sperm energy metabolism or NO concentration, suggesting a negligible effect of this wavelength on the analyzed parameters. In contrast, both spermatozoa irradiated at 810 nm and those treated with other parameters at 450 nm showed a modulated increase in NO concentration, consistent with a progressive increase in cellular energy.

Irradiation at 635 nm highlights the existence of critical thresholds in regulating mitochondrial activity. Under this condition, the ATP/AMP ratio underwent a dose-dependent reduction, accompanied by a proportional increase in NO concentration. This increase ranged between 400% and 800% compared to controls, and between 40% and 150% compared to spermatozoa irradiated at 810 nm with 1 W (60 J), representing the experimental setup with the highest ATP production. The results agree with evidence suggesting that PBM can stimulate NO synthesis and its release from S-nitroso proteins and hemoproteins [52,99,100,101,102]. Although nitric oxide synthase (NOS) is not considered a primary photoacceptor, it contains a heme group that can absorb light and modulate NO availability, a mechanism reminiscent of other heme proteins [103]. Furthermore, the presence of flavins in the reductase domain of NOS adds a layer of light sensitivity. The modulated increase in NO could thus improve the efficiency of OxPhos by acting on substrate metabolism or result from NO release due to photo-energization of mitochondrial complexes, which competitively inhibits oxygen by binding to the heme a_3_ and Cu_B groups of Complex IV [104,105], slowing ATP production efficiency. However, at high concentrations, NO can strongly inhibit Complex IV, leading to mitochondrial metabolic collapse and a drastic reduction in ATP production [104,105].

Donnelly et al. [106] observed that spermatozoa of normozoospermic men produce more NO under basal conditions than asthenozoospermic ones, but the difference is not statistically significant. However, following stimulation with the calcium ionophore A23187, asthenozoospermic spermatozoa show a more substantial increase in NO production compared to normozoospermic ones (+203.85% vs. +143.75% compared to baseline levels), suggesting a possible alteration in NO regulation. Balercia et al. [107] report opposite results, highlighting that asthenozoospermic spermatozoa have significantly higher NO levels than normozoospermic ones. Moreover, they describe a significant negative correlation between NO and motility, suggesting that high NO concentrations may compromise sperm functionality.

Our data on sperm energetics align with the study by Zhang and Zheng [97], which resolves the contradictions between previous studies by introducing the concept of a biphasic effect of NO on sperm motility. Medium-low concentrations of NO stimulate cGMP production, improving sperm motility. In contrast, high NO concentrations exert a toxic effect, likely due to the formation of peroxynitrite (ONOO^−^), resulting from the reaction between NO and superoxide (O_2_^−^).

Even in the case of NO at 60 min, a reduction of the initially observed values is noted. The correlation between NO production and energy metabolism is no longer evident with deeper NIR irradiation, particularly at 940 and 1064 nm.

In samples irradiated at 1064 nm, NO concentration appears significantly higher than in controls and comparable to that observed with irradiation at 450 nm and 810 nm. However, despite the modulated NO production, cellular energetics are strongly inhibited at higher parameters, as observed with irradiation at 635 nm.

Therefore, while at visible wavelengths and 810 nm, the primary mediators of NO production modulation and sperm energetics appear to be mitochondrial metabolism, photoreceptors, and enzymatic hemoproteins, at 940–1064 nm, additional factors such as water and lipids come into play. These elements introduce alternative regulatory mechanisms that can influence sperm mitochondrial bioenergetics and modulate the interaction between light and cells in a manner distinct from other wavelengths. Notably, the higher absorption of light by lipids at 1064 nm compared to 940 nm may cause the harmful effect observed at 41 °C after 1064 nm irradiation, compared to 44 °C at 940 nm (Appendix A). This assumption is consistent with the temperature analysis (Appendix A), which indicates that the increase from 37.0 °C to 42.0 °C at 450 nm, likely due to the presence of potential chromophores [108,109], did not inhibit sperm energetic metabolism. On the other hand, the slight increase in temperature up to 39.5 °C at 635 nm is associated with the most damaging effect on sperm energetic status.

In the scientific literature, mitochondrial dysfunction and a significant increase in NO production are associated with stress and damage in spermatozoa [7,10,19,107,110]. Such pathological conditions seem to originate mainly from altered environments, typical of patients with specific pathologies, medical treatments, or incorrect lifestyles [111,112]. Although asthenozoospermic spermatozoa exhibit higher lipid peroxidation levels than normal ones, our data and previous studies have shown that PBM does not exacerbate these conditions [21]. It is plausible that the potential adverse effects of irradiating ejaculated spermatozoa maintained in their seminal plasma are mitigated by the high antioxidant capacity of the seminal plasma itself [113,114]. However, it is essential to consider that, at low concentrations, most of the inhibition of Complex IV by NO is reversible under normoxic conditions. In contrast, at high concentrations, peroxynitrite formation can favor the persistence of enzymatic blockades for extended periods (like irreversible mode) [105].

The demonstration that PBM with wavelengths in the visible and NIR range positively modulates the energy metabolism of asthenozoospermic spermatozoa without harmful oxidative stress may allow potential applications in clinical practice as an asthenozoospermia therapy. Mammalian sperm uses ATP for several important functions, including motility, capacitation, hyperactivation, acrosomal reaction, and oocyte penetration. First, one possible translational application may be in fertility clinics performing intrauterine inseminations. In fact, the percentage of progressively motile sperm is one of the most critical prognostic indicators of fertilization. Thus, boosting the energetic status of spermatozoa by PBM can improve the chance of pregnancy and could be helpful in cases of moderate male infertility to encourage the use of intrauterine insemination as a cost-effective first-line treatment. Second, a laser-induced improvement in sperm motility may help embryologists to select viable spermatozoa in patients with immotile sperm during oocyte intracytoplasmic sperm injection (ICSI). A method of testing sperm viability is to induce sperm motility by increasing cyclic adenosine monophosphate (cAMP) levels by treating a semen sample with phosphodiesterase inhibitors, such as theophylline and pentoxifylline [115]. Although pentoxifylline is a popular in vitro sperm motility enhancer in many embryology laboratories, its application is not commonly approved in clinical practice because the consequences of pentoxifylline treatments in terms of potential damage to sperm DNA, oocytes, embryos, and newborns are contradictory [116]. In this context, stimulating sperm motility by PBM (i.e., at 810 nm) could advance sperm selection in ART procedures. A third possible translational application is in sperm preparation protocols for frozen and thawed semen, especially in fertility preservation programs for cancer patients before therapies. Despite its widespread use, the cryopreservation technique can cause damage to sperm through oxidative stress, which induces lipid peroxidation, DNA fragmentation, and apoptosis [117]. The harmful side effects of the freezing and thawing process may be countered by PBM preconditioning of fresh semen before the cryopreservation procedure, potentially resulting in increased mitochondrial membrane potential and decreased levels of intracellular ROS and lipid peroxidation.

Of course, ethical concerns must be resolved before the use of PBM to sperm in assisted reproductive technologies (ART), whose ultimate goal is a healthy baby. There is currently no published information on the potential adverse effects of PBM on sperm for the developing human embryo, making further research necessary.

## 5. Conclusions

In this study, we demonstrated that PBM with wavelengths in the visible and NIR ranges, applied for 60 s, can significantly modulate the energy metabolism of asthenozoospermic spermatozoa, with rapid and notably lasting effects up to 60 min post-treatment. Specifically, irradiation at 810 nm proved the most effective in improving ATP production and the ATP/AMP ratio. In contrast, irradiation at 635 nm impaired energy metabolism.

An important finding was the identification of a correlation between NO production and the energy metabolism of spermatozoa.

Our results suggest that PBM mechanisms primarily involve mitochondrial photoreceptors and potentially the heme groups and flavins of NOS, which mediate electronic transitions, enhance OxPhos efficiency, and improve enzymatic activity. At longer wavelengths (940 nm and 1064 nm), photothermal mechanisms may negatively influence membrane fluidity and mitochondrial functionality in a complex manner, inconsistent with what is observed at shorter wavelengths. Regardless of the spermatozoa’s treatment, PBM appeared safe with respect to oxidative stress damage to lipids and DNA.

## Figures and Tables

**Figure 1 biology-14-00491-f001:**
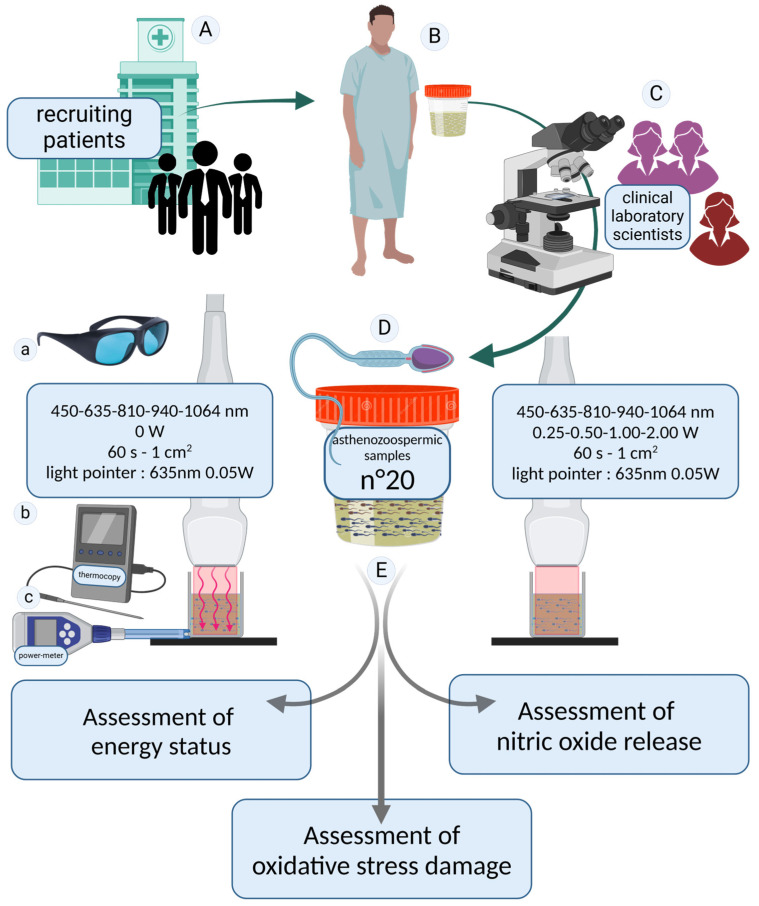
Study design. Patients were recruited at the SSD Physiopathology of Human Reproduction, IRCCS Ospedale Policlinico San Martino, Genoa, Italy (**A**). Seminal fluid samples were donated (**B**) and analyzed according to the inclusion and exclusion criteria (**C**). To ensure maximum reliability of the results, the diagnosis of asthenozoospermia was confirmed through two independent blinded evaluations conducted by healthcare professionals specialized in seminology. In cases of discrepancies between the two assessments, a third experienced operator performed an additional analysis to establish a definitive diagnosis. Twenty asthenozoospermic semen samples were included in our study (**D**). The asthenozoospermic samples were divided into aliquots and irradiated with photobiomodulation therapies at 450, 635, 810, 940, and 1064 nm for 60 *s*. Samples irradiated with the laser set at 0 W were considered as controls. A 635 nm guide light emitting negligible energy was used to set up the irradiation. All irradiations were performed while wearing laser safety goggles compliant with European regulations (**a**). The consistency of the irradiated values was monitored using a power meter (**b**), and temperature variations were measured using a thermocouple (**c**). To investigate the effects of photobiomodulation treatment (**E**), the energetic state of the spermatozoa, oxidative stress damage, and nitric oxide release were evaluated. The image was created at https://BioRender.com by A.A (accessed on 30 April 2025).

**Figure 2 biology-14-00491-f002:**
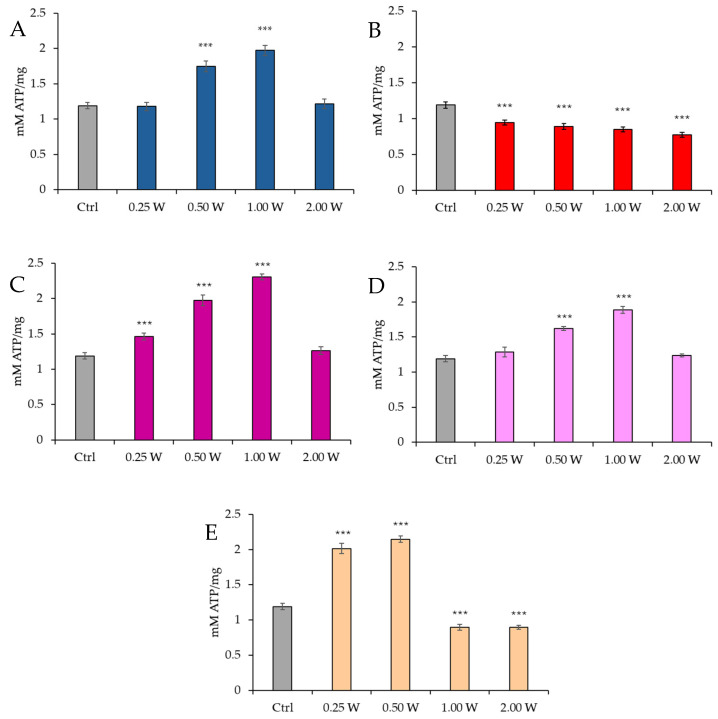
Effect of photobiomodulation on ATP concentration 10 min after treatment of 20 semen samples at different wavelengths (450 nm (**A**); 635 nm (**B**); 810 nm (**C**); 940 nm (**D**); and 1064 nm (**E**)) and settings of 0.25 W (15 J), 0.50 W (30 J), 1.00 W (60 J), and 2.00 W (120 J) for 60 s. ***: *p* < 0.001 vs. the control. All the graphs show the mean and standard deviation (error bar: 95% confidence interval for mean) of single measurements.

**Figure 3 biology-14-00491-f003:**
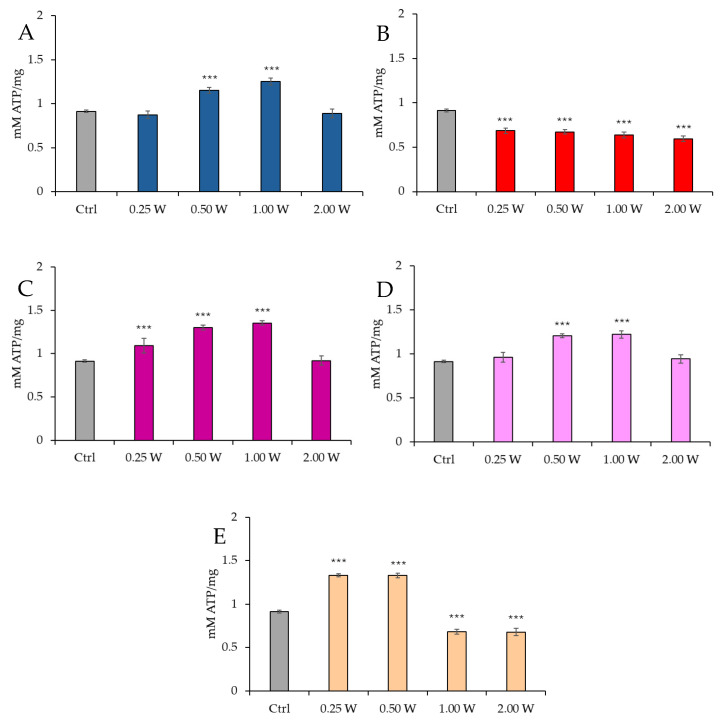
Effect of photobiomodulation on ATP concentration 60 min after treatment of 20 semen samples at different wavelengths (450 nm (**A**); 635 nm (**B**); 810 nm (**C**); 940 nm (**D**); and 1064 nm (**E**)) and settings of 0.25 W (15 J), 0.50 W (30 J), 1.00 W (60 J), and 2.00 W (120 J) for 60 s. All the graphs show the mean and standard deviation (error bar: 95% confidence interval for mean) of single measurements. ***: *p* < 0.001 vs. the control.

**Figure 4 biology-14-00491-f004:**
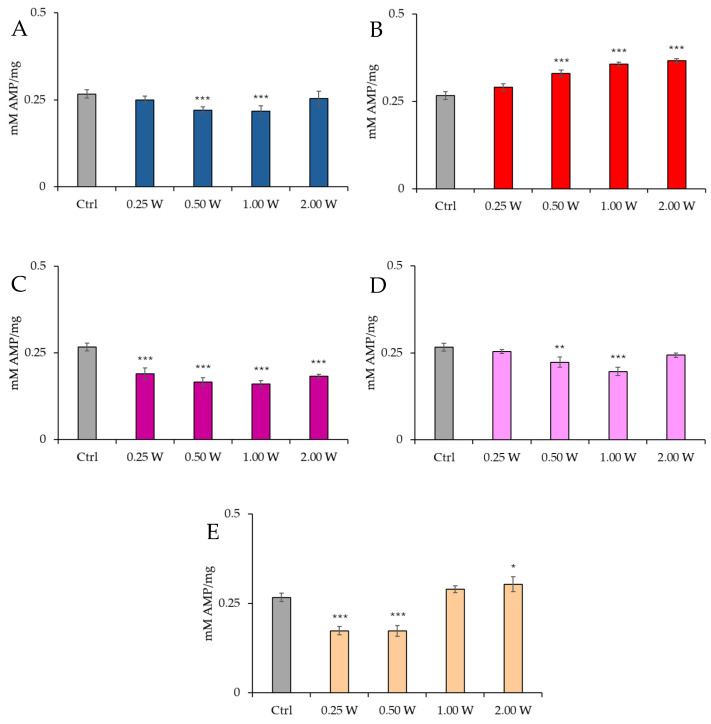
Effect of photobiomodulation on AMP concentration 10 min after treatment of 20 semen samples at different wavelengths (450 nm (**A**); 635 nm (**B**); 810 nm (**C**); 940 nm (**D**); 1064 nm (**E**)) settings of 0.25 W (15 J), 0.50 W (30 J), 1.00 W (60 J), and 2.00 W (120 J) for 60 s. All the graphs show the mean and standard deviation (error bar: 95% confidence interval for mean) of single measurements. *: *p* < 0.05; **: *p* < 0.01; ***: *p* < 0.001 vs. the control.

**Figure 5 biology-14-00491-f005:**
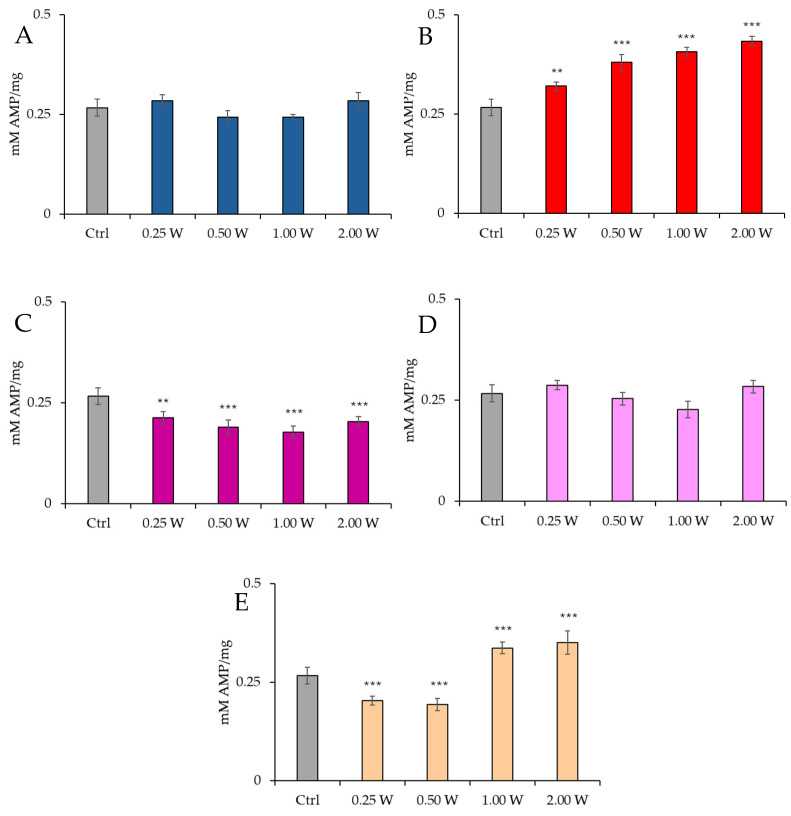
Effect of photobiomodulation on AMP concentration 60 min after treatment of 20 semen samples at different wavelengths (450 nm (**A**); 635 nm (**B**); 810 nm (**C**); 940 nm (**D**); and 1064 nm (**E**)) and settings of 0.25 W (15 J), 0.50 W (30 J), 1.00 W (60 J), and 2.00 W (120 J) for 60 s. All the graphs show the mean and standard deviation (error bar: 95% confidence interval for mean) of single measurements. **: *p* < 0.01; ***: *p* < 0.001 vs. the control.

**Figure 6 biology-14-00491-f006:**
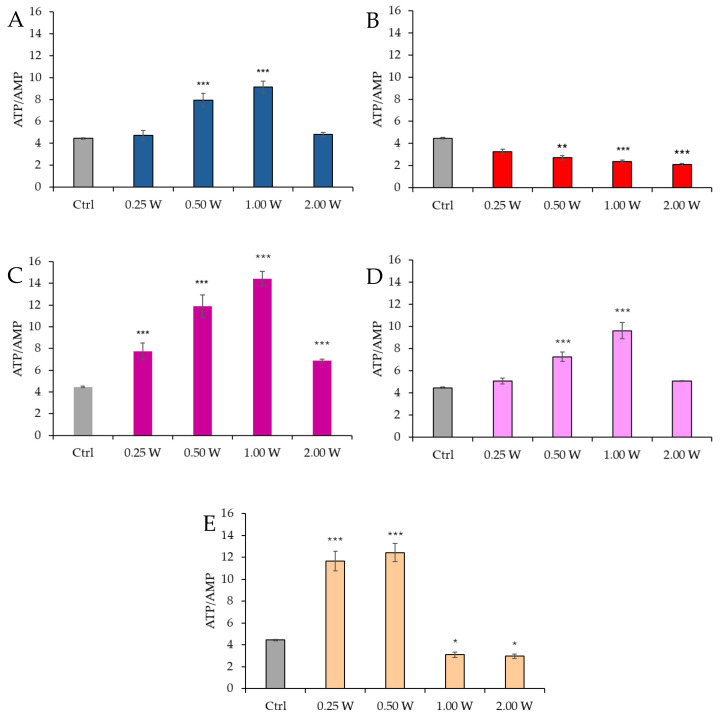
Effect of photobiomodulation on ATP/AMP ratio 10 min after treatment of 20 semen samples at different wavelengths (450 nm (**A**); 635 nm (**B**); 810 nm (**C**); 940 nm (**D**); and 1064 nm (**E**)) and settings of 0.25 W (15 J), 0.50 W (30 J), 1.00 W (60 J), and 2.00 W (120 J) for 60 s. All the graphs show the mean and standard deviation (error bar: 95% confidence interval for mean) of single measurements. *: *p* < 0.05; **: *p* < 0.01; ***: *p* < 0.001 vs. the control.

**Figure 7 biology-14-00491-f007:**
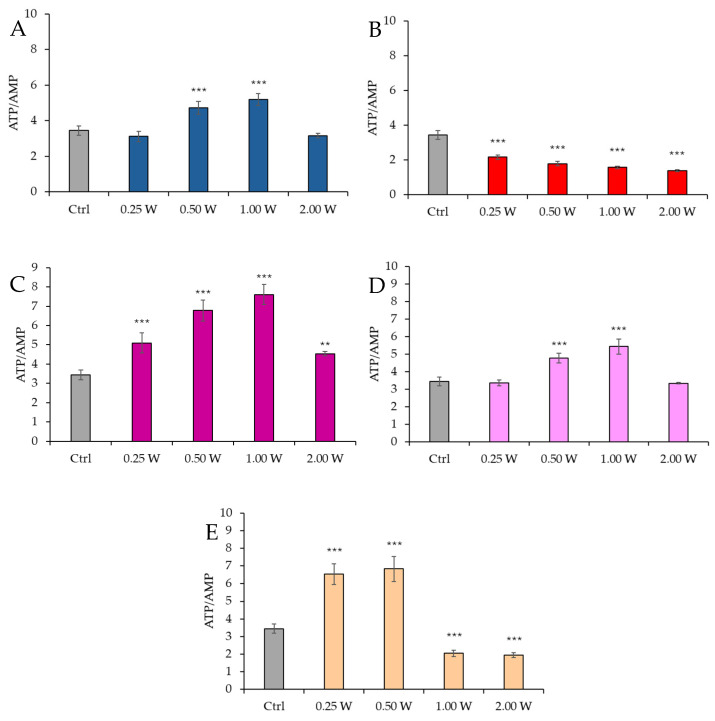
Effect of photobiomodulation on ATP/AMP ratio 60 min after treatment of 20 semen samples at different wavelengths (450 nm (**A**); 635 nm (**B**); 810 nm (**C**); 940 nm (**D**); and 1064 nm (**E**)) and settings of 0.25 W (15 J), 0.50 W (30 J), 1.00 W (60 J), and 2.00 W (120 J) for 60 s. All the graphs show the mean and standard deviation (error bar: 95% confidence interval for mean) of single measurements. ***: *p* < 0.001, **: *p* < 0.01 vs. the control.

**Figure 8 biology-14-00491-f008:**
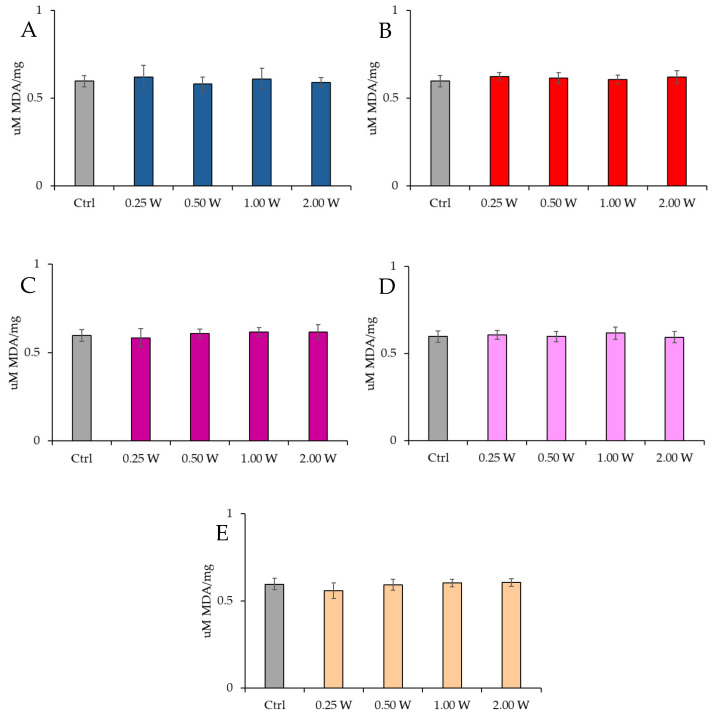
Effect of photobiomodulation on MDA concentration 10 min after treatment of 20 semen samples at different wavelengths (450 nm (**A**); 635 nm (**B**); 810 nm (**C**); 940 nm (**D**); and 1064 nm (**E**)) and settings of 0.25 W (15 J), 0.50 W (30 J), 1.00 W (60 J), and 2.00 W (120 J) for 60 s. No significant differences were observed. All the graphs show the mean and standard deviation (error bar: 95% confidence interval for mean) of single measurements.

**Figure 9 biology-14-00491-f009:**
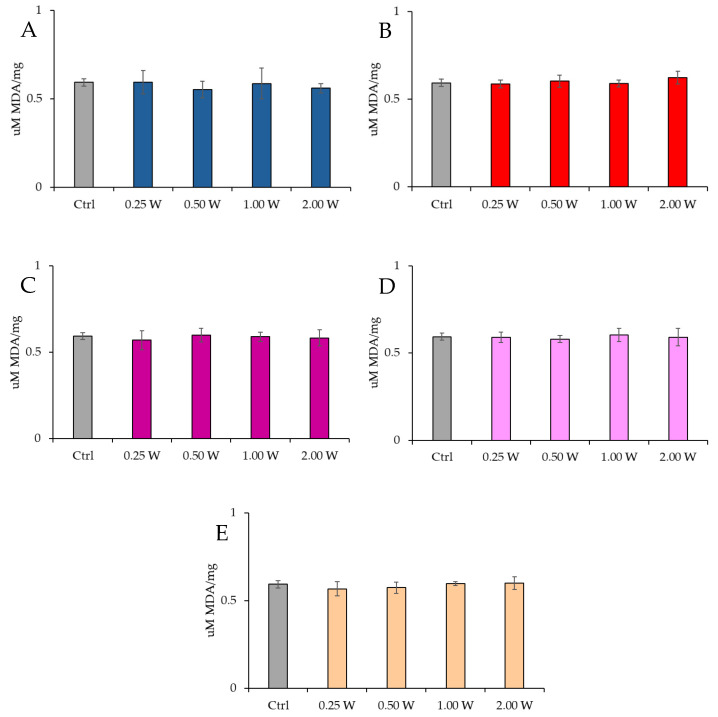
Effect of photobiomodulation on MDA concentration 60 min after treatment of 20 semen samples at different wavelengths (450 nm (**A**); 635 nm (**B**); 810 nm (**C**); 940 nm (**D**); and 1064 nm (**E**)) and settings of 0.25 W (15 J), 0.50 W (30 J), 1.00 W (60 J), and 2.00 W (120 J) for 60 s. No significant differences were observed. All the graphs show the mean and standard deviation (error bar: 95% confidence interval for mean) of single measurements.

**Figure 10 biology-14-00491-f010:**
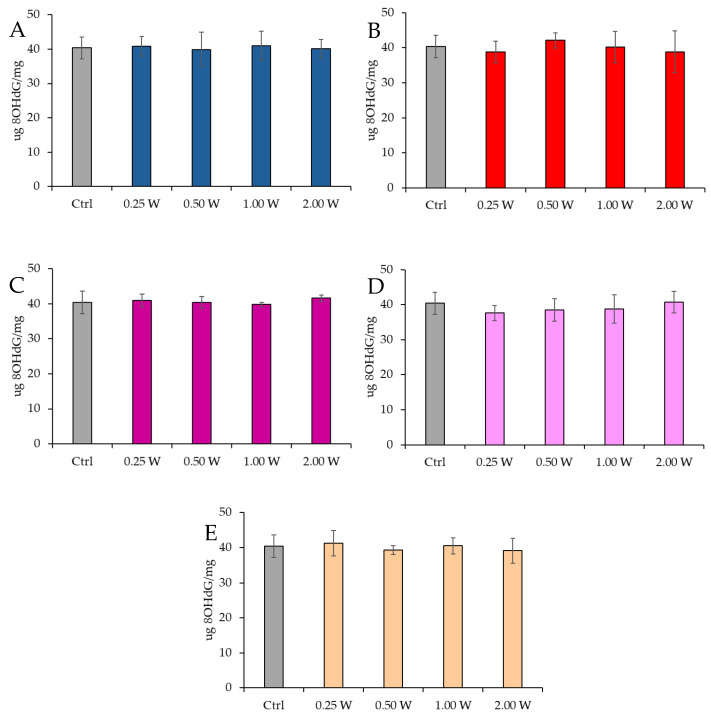
Effect of photobiomodulation on 8-OHdG concentration 10 min after treatment of 20 semen samples at different wavelengths (450 nm (**A**); 635 nm (**B**); 810 nm (**C**); 940 nm (**D**); and 1064 nm (**E**)) and settings of 0.25 W (15 J), 0.50 W (30 J), 1.00 W (60 J), and 2.00 W (120 J) for 60 s. No significant differences were observed. All the graphs show the mean and standard deviation (error bar: 95% confidence interval for mean) of single measurements.

**Figure 11 biology-14-00491-f011:**
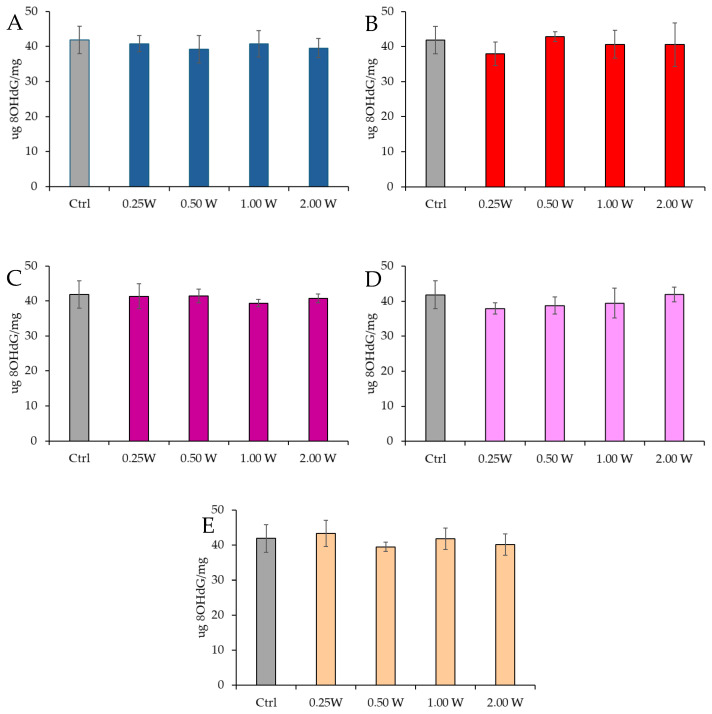
Effect of photobiomodulation on 8-OHdG concentration 60 min after treatment of 20 semen samples at different wavelengths (450 nm (**A**); 635 nm (**B**); 810 nm (**C**); 940 nm (**D**); and 1064 nm (**E**)) and settings of 0.25 W (15 J), 0.50 W (30 J), 1.00 W (60 J), and 2.00 W (120 J) for 60 s. No significant differences were observed. All the graphs show the mean and standard deviation (error bar: 95% confidence interval for mean) of single measurements.

**Figure 12 biology-14-00491-f012:**
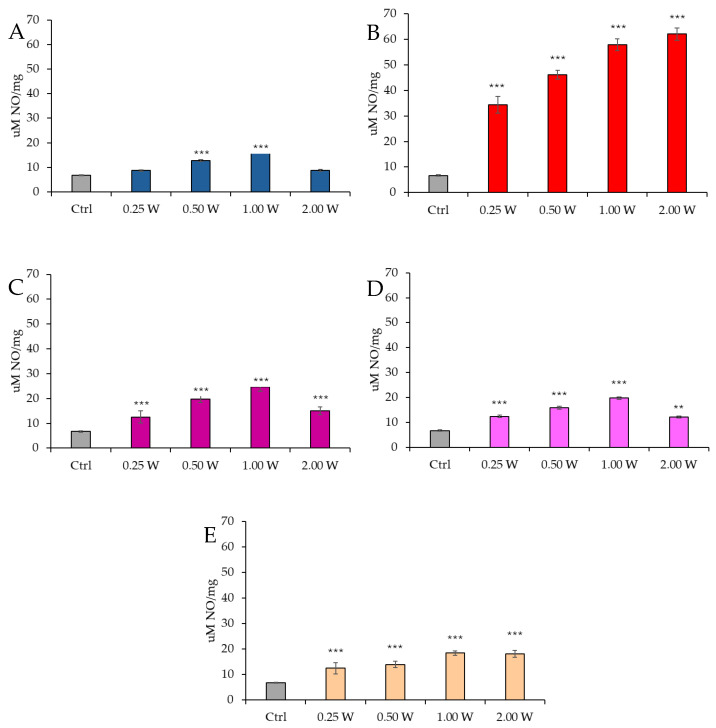
Effect of photobiomodulation on NO production evaluated as nitrite concentration 10 min after treatment of 20 semen samples at different wavelengths (450 nm (**A**); 635 nm (**B**); 810 nm (**C**); 940 nm (**D**); and 1064 nm (**E**)) and settings of 0.25 W (15 J), 0.50 W (30 J), 1.00 W (60 J), and 2.00 W (120 J) for 60 s. All the graphs show the mean and standard deviation (error bar: 95% confidence interval for mean) of single measurements. **. *p* < 0.01; ***: *p* < 0.001 vs. the control.

**Figure 13 biology-14-00491-f013:**
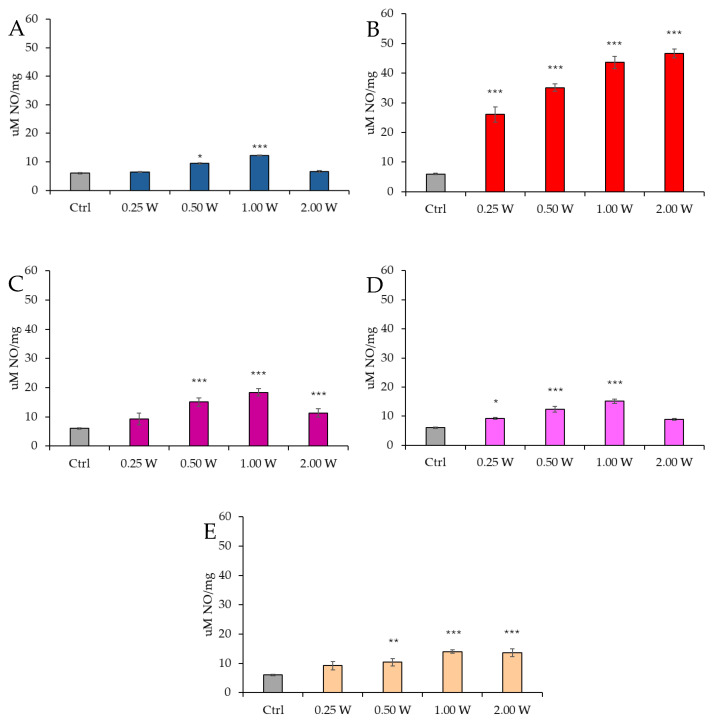
Effect of photobiomodulation on NO production evaluated as nitrite concentration 60 min after treatment of 20 semen samples at different wavelengths (450 nm (**A**); 635 nm (**B**); 810 nm (**C**); 940 nm (**D**); and 1064 nm (**E**)) and settings of 0.25 W (15 J), 0.50 W (30 J), 1.00 W (60 J), and 2.00 W (120 J) for 60 s. All the graphs show the mean and standard deviation (error bar: 95% confidence interval for mean) of single measurements. *: *p* < 0.05; **: *p* < 0.01; ***: *p* < 0.001 vs. the control.

**Table 1 biology-14-00491-t001:** Demographic data of the 20 patients enrolled in the study and their semen parameters before the experiments.

Characteristics	Mean ± SEM
Age (years)	35.74 ± 2.06
Volume (mL)	3.51 ± 0.26
Sperm concentration (10^6^/mL)	34.74 ± 4.73
Normal morphological sperm (%)	3.32 ± 0.57
Total motility (%)	34.10 ± 1.31
Progressive motility (%)	28.10 ± 1.03
Sperm vitality (%)	54.26 ± 1.56

**Table 2 biology-14-00491-t002:** Description of the equipment used for photobiomodulation irradiation and the parameters delivered to the spermatozoa.

Laser Model	ENEA Trio diode laser system (Garda Laser S.A.S., Verona, Italy) https://gardalaser.it/ (accessed on 30 April 2025)
Irradiation device	Flat-profiled handpiece [58]
Max. power in continuous wave mode	15 Watts (W)
Light pointer	635 nm red light at negligible power output, <0.5 mW
Irradiation parameters
Wavelengths	450 nm ± 10 nm	635 nm ± 10 nm	810 nm ± 10 nm	940 nm ± 10 nm	1064 nm ± 10 nm	

	Power	Time	Energy	Area	Power-Density	Fluence
0.25 W	60 s	15 J	1 cm^2^	0.25 W/cm^2^	15 J/cm^2^
0.50 W	60 s	30 J	1 cm^2^	0.50 W/cm^2^	30 J/cm^2^
1.00 W	60 s	60 J	1 cm^2^	1.00 W/cm^2^	60 J/cm^2^
2.00 W	60 s	120 J	1 cm^2^	2.00 W/cm^2^	120 J/cm^2^
Irradiation performed in continuous wave mode

## Data Availability

The data are available on request from the authors.

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
