# Peer review of "Efficacy and Safety of Visible and Near-Infrared Photobiomodulation Therapy on Astenospermic Human Sperm: Wavelength-Dependent Regulation of Nitric Oxide Levels and Mitochondrial Energetics"

_biology, 2025, doi:10.3390/biology14050491_

Round 1

Reviewer 1 Report

Comments and Suggestions for Authors

Thank you for the manuscript. It's well written in general. Here are two minor suggestions to improve it.

(1) Please mention the very large magnitude of NO increase from red light irradiation (compared to other wavelengths) in the abstract.

(2) Please consider citing this older paper that previously also noted that red light has more effect on NO release than near-infrared light:

----> Keszler et al. Wavelength-dependence of vasodilation and NO release from S-nitrosothiols and dinitrosyl iron complexes by far red/near infrared light (2018) <----

Author Response

Genoa, 24th April 2025

Dear Colleague, Reviewer 1,

We sincerely thank you for your time and effort in reviewing our manuscript. Your insightful comments and constructive suggestions have been invaluable in improving the quality and clarity of our work.

All revisions prompted by your observations have been carefully addressed and are highlighted in yellow in the main text for your convenience.

Below, we provide detailed, point-by-point responses to each of your comments.

Q1 Please mention the very large magnitude of NO increase from red light irradiation (compared to other wavelengths) in the abstract.

A1. We have added this information in the Abstract.

Q2. Please consider citing this older paper that previously also noted that red light has more effect on NO release than near-infrared light:

----> Keszler et al. Wavelength-dependence of vasodilation and NO release from S-nitrosothiols and dinitrosyl iron complexes by far red/near infrared light (2018) <----

A2. We have cited the paper by Keszler et al in the Introduction section as new reference #47

We hope that our modifications meet your expectations and reflect our appreciation of your valuable feedback.

With kind regards,
Andrea Amaroli
Professor, Department of Earth, Environmental and Life Sciences (DISTAV)
University of Genoa

Reviewer 2 Report

Comments and Suggestions for Authors

Interesting study about the impact of BPM on sperm function. The study design is sound also the explanations in Methods. 

However, in Discussion the authors should concentrate more on the thermal effects of their laser therapy. In the table 1S you see temperatures of 44 C which do not harm (940 nm), wheras at 1064 nm already 41 C is harmful. The same point and even more conflicting in 635 nm already beginning at 37 C. Here, the wavelength of light are potentially interfering with some enzymatic processes...

Minor point: please correct spacing especially before citations.

Author Response

Genoa, 24th April 2025

Dear Colleague, Reviewer 2,

We sincerely thank you for your time and effort in reviewing our manuscript. Your insightful comments and constructive suggestions have been invaluable in improving the quality and clarity of our work.

All revisions prompted by your observations have been carefully addressed and are highlighted in green in the main text for your convenience.

Below, we provide detailed, point-by-point responses to each of your comments.

Q1. However, in Discussion the authors should concentrate more on the thermal effects of their laser therapy. In the table 1S you see temperatures of 44 C which do not harm (940 nm), wheras at 1064 nm already 41 C is harmful. The same point and even more conflicting in 635 nm already beginning at 37 C. Here, the wavelength of light are potentially interfering with some enzymatic processes...

A1. We have detailed this issue in the Discussion section.

Q2. Minor point: please correct spacing especially before citations.

A2. Thanks to the Reviewer’s comment we have corrected the typing errors

We hope that our modifications meet your expectations and reflect our appreciation of your valuable feedback.

With kind regards,
Andrea Amaroli
Professor, Department of Earth, Environmental and Life Sciences (DISTAV)
University of Genoa

Reviewer 3 Report

Comments and Suggestions for Authors

A very interesting study with relevant contributions. We suggest addressing the following observations:

Was some care taken regarding the ambient light exposure of the samples before processing or before undergoing experimentation with different light ranges? Explain whether this ambient light prior to the experimental treatment may or may not alter the results.

Photobiomodulation treatments: Please specify the exposure time. Although 60 seconds is used in a table, when the experiment is initially described, several parameters are mentioned, but not the time; please add this.

In the statistical analysis, you mention that you used ANOVA, but not the post-hoc test. The ANOVA test is also a parametric measure for normally distributed data. Please specify how you determined the normality of the data. Likewise, with the sample size available, is it feasible to use ANOVA, or would a non-parametric test be better? Please explain in the statistics section.

Please specify in each graph and analysis how much data was included per group. Although the number of samples per group is mentioned, it is not stated whether they were single measurements or duplicates, etc.

The figures do not mention whether the mean and standard deviation are graphed, or what is being graphed.

Author Response

Genoa, 24th April 2025

Dear Colleague, Reviewer 3,

We sincerely thank you for your time and effort in reviewing our manuscript. Your insightful comments and constructive suggestions have been invaluable in improving the quality and clarity of our work.

All revisions prompted by your observations have been carefully addressed and are highlighted in blue in the main text for your convenience.

Below, we provide detailed, point-by-point responses to each of your comments.

Q1. Was some care taken regarding the ambient light exposure of the samples before processing or before undergoing experimentation with different light ranges? Explain whether this ambient light prior to the experimental treatment may or may not alter the results.

A1. We have detailed that to avoid potential biases, the experiments were performed without any artificial light in the “Technical Specifications of the Equipment Utilized for Spermatozoa Irradiation” paragraph of the Materials and Methods section. Indeed, the indirect natural light is not sufficient to induce the photoenergization of chromophores involved in mitochondrial metabolism.

Q2. Photobiomodulation treatments: Please specify the exposure time. Although 60 seconds is used in a table, when the experiment is initially described, several parameters are mentioned, but not the time; please add this.

A2. We have added the exposure time at 60 seconds in the “Patient Recruitment and Experimental Design” paragraph of the Materials and Methods section.

Q3. In the statistical analysis, you mention that you used ANOVA, but not the post-hoc test. The ANOVA test is also a parametric measure for normally distributed data. Please specify how you determined the normality of the data. Likewise, with the sample size available, is it feasible to use ANOVA, or would a non-parametric test be better? Please explain in the statistics section.

A3. Thanks to the Reviewer’s comment, we have better explained the statistical approach we used (see “Statistical Analysis” paragraph in the Materials and Methods section).

Q4. Please specify in each graph and analysis how much data was included per group. Although the number of samples per group is mentioned, it is not stated whether they were single measurements or duplicates, etc. The figures do not mention whether the mean and standard deviation are graphed, or what is being graphed.

A4. In the “Statistical Analysis” paragraph in the Materials and Methods section, we have specified that for each analysis we collected single measurements. Moreover, in the Legends of Figures 2-13, we added the number of semen samples (n=20) analyzed and that each graph shows the mean and standard deviation of single measurements.

We hope that our modifications meet your expectations and reflect our appreciation of your valuable feedback.

With kind regards,
Andrea Amaroli
Professor, Department of Earth, Environmental and Life Sciences (DISTAV)
University of Genoa

Reviewer 4 Report

Comments and Suggestions for Authors

 Title

The title is comprehensive and clearly reflects the study's focus on photobiomodulation therapy (PBM) and its effects on asthenospermic sperm.

Abstract

The abstract effectively summarizes the study's objectives, methods, and key findings. It provides a clear rationale for investigating PBM in asthenozoospermia and succinctly presents the results.

Introduction

The introduction provides a thorough background on male infertility, mitochondrial dysfunction, and nitric oxide's role in sperm motility. It effectively links prior research to the study's rationale. However, the section would benefit from a clearer hypothesis or research gap statement early on. The references are well-chosen but could include more recent studies (e.g., post-2020) to reinforce the novelty of the work.

Materials and Methods

The methodology is detailed and well-structured, with clear descriptions of patient recruitment, irradiation parameters, and assays. The inclusion of temperature monitoring and power verification adds rigor. However, some technical details (e.g., laser specifications) could be moved to a supplementary table to improve readability. The major question is why the authors did not evaluate other routine sperm parameters that can be evaluated before and after the experiment and compare them among different groups, such as morphology, motility, viability, and also DNA fragmentation. For completing this part, it is suggested to complete data otherwise if these data are not available, only add the demographic data of the included patients that the authors used for the experiments and their sperm characteristics. It is suggested to refer to the study below to complete this part.

Evaluation of supplementation of cryopreservation medium with gallic acid as an antioxidant in quality of post‐thaw human spermatozoa

Results

The results are presented systematically, with clear figures and statistical annotations. The data on ATP/AMP ratios, oxidative stress markers, and NO levels are well-organized. However, the text occasionally repeats information already visible in the figures (e.g., "At 810 nm, all irradiations produced a significant decrease in AMP concentration"). Some subsections could be condensed for brevity. The temporal comparisons (10 vs. 60 minutes) are a strength, but the biological significance of the decline in ATP/AMP ratios over time could be discussed more in the results or reserved for the discussion.

Discussion

The discussion effectively interprets the results in the context of mitochondrial function, NO dynamics, and prior PBM studies. The biphasic response of NO and the differential effects of wavelengths are well-explained. However, the section could be more focused—some paragraphs delve into tangential mechanisms (e.g., water/lipid interactions) without clearly tying them back to the central findings. The clinical implications are underdeveloped; explicitly stating how these results could translate to fertility treatments would strengthen the impact. The safety data (no oxidative stress damage) is a key strength but could be highlighted earlier.

Author Response

Genoa, 24th April 2025

Dear Colleague, Reviewer 4,

We sincerely thank you for your time and effort in reviewing our manuscript. Your insightful comments and constructive suggestions have been invaluable in improving the quality and clarity of our work.

All revisions prompted by your observations have been carefully addressed and are highlighted in fuchsia in the main text for your convenience.

Below, we provide detailed, point-by-point responses to each of your comments.

Introduction

The introduction provides a thorough background on male infertility, mitochondrial dysfunction, and nitric oxide's role in sperm motility. It effectively links prior research to the study's rationale.

Q1. However, the section would benefit from a clearer hypothesis or research gap statement early on.

A1. We have revised the last paragraph of the Introduction to clearly explain the scope of our study taking into account the gap of information available in the literature.

Q2. The references are well-chosen but could include more recent studies (e.g., post-2020) to reinforce the novelty of the work.

A2. We have introduced some references from the last 4 years in the introduction

Materials and Methods

The methodology is detailed and well-structured, with clear descriptions of patient recruitment, irradiation parameters, and assays. The inclusion of temperature monitoring and power verification adds rigor. However, some technical details (e.g., laser specifications) could be moved to a supplementary table to improve readability.

Q3. The major question is why the authors did not evaluate other routine sperm parameters that can be evaluated before and after the experiment and compare them among different groups, such as morphology, motility, viability, and also DNA fragmentation. For completing this part, it is suggested to complete data otherwise if these data are not available, only add the demographic data of the included patients that the authors used for the experiments and their sperm characteristics.

A3. We did not evaluate sperm parameters after experiments since the aim of this study was to provide scientific evidence on the efficacy and safety of PBM for treating asthenozoospermia focusing on the metabolic aspects of sperm. As required by the Reviewer, we added a Table (new Table 1) to summarize the demographic data of the 20 patients enrolled in the study and their semen parameters before the experiments.

Results

The results are presented systematically, with clear figures and statistical annotations. The data on ATP/AMP ratios, oxidative stress markers, and NO levels are well-organized.

Q4. The temporal comparisons (10 vs. 60 minutes) are a strength, but the biological significance of the decline in ATP/AMP ratios over time could be discussed more in the results or reserved for the discussion.

A4. We completely agree with the Reviewer that the temporal comparison of ATP/AMP decline is an interesting finding. In fact, we have highlighted it in the “Energy Status (ATP/AMP Ratio) Evaluation in Spermatozoa in Response to Photobiomodulation” of the Results section and the Discussion.

Discussion

The discussion effectively interprets the results in the context of mitochondrial function, NO dynamics, and prior PBM studies. The biphasic response of NO and the differential effects of wavelengths are well-explained. Q4. However, the section could be more focused—some paragraphs delve into tangential mechanisms (e.g., water/lipid interactions) without clearly tying them back to the central findings.

Q5. The clinical implications are underdeveloped; explicitly stating how these results could translate to fertility treatments would strengthen the impact.

A5. We thank the Reviewer for this comment. We strengthen the putative translational application of PBM in the clinic by discussing its possible use in fertility treatments at the end of the Discussion section

Q6. The safety data (no oxidative stress damage) is a key strength but could be highlighted earlier.

A6. We thank the Reviewer for the suggestion. In the revised version, the safety of irradiation has been mentioned in the third paragraph of the discussion that summarized the obtained results.

We hope that our modifications meet your expectations and reflect our appreciation of your valuable feedback.

With kind regards,
Andrea Amaroli
Professor, Department of Earth, Environmental and Life Sciences (DISTAV)
University of Genoa

Round 2

Reviewer 3 Report

Comments and Suggestions for Authors

the changes have been made. the manuscript may be accepted for publication

Author Response

We appreciate the positive comments of the Reviewer